# [ICLR26] TOTAL VARIATION BY PINN

## ABSTRACT

PINN approximation for solutions of non-linear equations is much more difficult than the linear analogue. This is due to the non-differentiability of the solution at some points and/or non-differentibility of the coefficients of the PDE. We develop a PINN approach for $\Gamma$-convergence inspired TV regularization à la Chambolle and Lions. We use smooth approximations of the TV functional that makes the problem differentiable at all points. Our coordinate-based neural network representation enables gradient computation while maintaining the continuous PDE formulation. We train the network through increasingly less smooth approximations, gradually approaching the original TV solution. We next extend our results from a single image and a single PDE to a class of images. This is done via operator learning that maps any initial image to its TV solution where a single network learns the denoising operator across multiple images. Experiments on 2D and 3D data demonstrate that our method achieves competitive denoising quality with classical TV solvers.

## 1 INTRODUCTION

Total variation (TV) regularization is fundamental for edge-preserving image denoising. The original TV problem minimizes

$$E(u) = \frac{1}{2}\|u - u_0\|_{L^2}^2 + \lambda \int_\Omega |\nabla u| dx \tag{1}$$

However, minimizing sequences in the natural space lack convergence guarantees, necessitating relaxation to the space of bounded variation (BV). This relaxed functional

$$\bar{E}(u) = \frac{1}{2}\|u - u_0\|_{L^2}^2 + \lambda TV(u) \tag{2}$$

while theoretically sound, involves measure-valued derivatives that are computationally challenging.

We solve the relaxed TV problem using physics-informed neural networks (PINNs) through a sequence of smooth approximations. Following Chambolle & Lions (1997), we replace the non-smooth functional with a decreasing sequence of smooth functionals

$$E_\varepsilon(u) = \frac{1}{2}\|u - u_0\|_{L^2}^2 + \lambda \int_\Omega \phi_\varepsilon(|\nabla u|) dx \tag{3}$$

that converge to the relaxed TV functional $\bar{E}$ in BV. Each $E_\varepsilon$ has a unique minimizer in $W^{1,2}$, and these minimizers converge in $L^1$ to the minimizer of $\bar{E}$.

Our approach transforms this framework into a neural network method. For each smooth functional $E_\varepsilon$, we minimize the corresponding Euler-Lagrange equation using a PINN Raissi et al. (2019). The smoothness enables gradient computation through automatic differentiation. This is a key advantage over finite difference methods used in classical solvers and recent neural approaches like DeepTV Langer & Behnamian (2024). We progressively reduce $\varepsilon$ during training to approach the TV solution. We extend this framework to operator learning using Lu et al. (2021) architecture , enabling a single network to learn the denoising operator across multiple images. Our experiments on 2D and 3D data, under both low and high noise conditions, demonstrate competitive performance with classical TV methods. **Contributions:** Our primary contribution is implementing the smooth approximation strategy for TV regularization using PINNs, inspired by Chambolle & Lions (1997),

demonstrating successful application in both 2D and 3D volumetric denoising. Additionally, we develop an operator learning framework that enables TV denoising across multiple images simultaneously through a single trained network.

## 2 RELATED WORK

Classical TV methods handle the non-differentiability of $|\nabla u|$ through various approaches. Chambolle Chambolle (2004) introduced a dual projection algorithm that avoids direct computation of the singular gradient term. Chambolle and Pock Chambolle & Pock (2011) extended this to primal-dual methods. Most relevant to our work, Chambolle and Lions Chambolle & Lions (1997) proved that smooth approximations $\phi_\varepsilon(|\nabla u|)$ of the TV functional $\Gamma$-converge to the relaxed TV solution as $\varepsilon \to 0$.

Recent neural approaches address TV-related problems. Grossmann et al. Grossmann et al. (2022) learn the TV flow PDE $\partial_t u = \nabla \cdot (\nabla u / |\nabla u|)$ using a network that takes the noisy image $u_0$ and time $t$ as inputs, outputting the denoised image $u(t)$ and a flux field. They compute spatial derivatives using finite differences on the network outputs.

Langer Langer & Behnamian (2024) establishes $\Gamma$-convergence for ReLU networks approximating TV minimizers, with convergence properties related to network width and weight bounds. They use coordinate-based networks but compute gradients through finite differences.

Our approach implements the Chambolle-Lions framework directly using coordinate-based PINNs with smooth activations. This enables automatic differentiation for all gradient computations, avoiding finite differences entirely. We solve the classical ROF functional through progressive $\varepsilon$-reduction, maintaining the theoretical $\Gamma$-convergence guarantees while extending to 3D applications.

## 3 BACKGROUND

### 3.1 IMAGE DENOISING AND TOTAL VARIATION

Image reconstruction is a fundamental inverse problem where the observed degraded image $u_0$ relates to the true image $u$ through:

$$u_0 = Ru + \eta \tag{4}$$

where $R : L^2(\Omega) \to L^2(\Omega)$ models degradation (blur, downsampling) and $\eta$ represents noise.

Direct inversion is ill-posed small noise perturbations cause large reconstruction errors. Total variation (TV) regularization Rudin et al. (1992) stabilizes this by exploiting the image structure, where natural images contain piecewise smooth regions separated by edges. The TV-regularized problem minimizes:

$$E(u) = \frac{1}{2} \int_\Omega |Ru - u_0|^2 \, dx + \lambda \int_\Omega |\nabla u| \, dx \tag{5}$$

where $\lambda > 0$ balances fidelity against smoothness.

### 3.2 MATHEMATICAL FRAMEWORK AND RELAXATION

The natural space for image reconstruction with TV regularization is $V = \{u \in L^2(\Omega) : \nabla u \in L^1(\Omega)\}$, but this space is non-reflexive and lacks the sequential compactness necessary for the direct method. We therefore work in $BV(\Omega)$, which retains boundedness of minimizing sequences and provides sequential weak* compactness. However, the extended functionalfrom $V$ to $BV$ still lacks weak* lower semicontinuity, requiring further regularization. This necessitates working with the relaxed functional:

$$\bar{E}(u) = \frac{1}{2} \int_\Omega |Ru - u_0|^2 \, dx + \lambda TV(u) \tag{6}$$

where the total variation for $u \in BV(\Omega)$ decomposes as:

$$TV(u) = \int_\Omega |\nabla u| dx + \int_{S_u} |u^+ - u^-| d\mathcal{H}^{d-1} + \int_{\Omega \setminus S_u} |Cu| dx \tag{7}$$

comprising the smooth part, jumps across discontinuities $S_u$, and the Cantor part $Cu$. This relaxation preserves the variational structure while gaining the analytical properties needed for existence theory. Minimizers of $E$ in $V$ remain minimizers of $\bar{E}$ in $BV$, and minimizing sequences now converge in the weak* topology to actual minimizers. Most importantly, $\bar{E}$ achieves lower semicontinuity, enabling the direct method to guarantee existence in BV.

### 3.3 NUMERICAL SOLUTION VIA SMOOTH APPROXIMATIONS

Direct optimization of $\bar{E}$ is challenging because it involves measure-valued terms. Chambolle and Lions Chambolle & Lions (1997) addressed this by constructing smooth approximations. They replace the non-smooth absolute value with a smooth function $\phi_\varepsilon$:

$$\phi_\varepsilon(s) = \begin{cases} \frac{s^2}{2\varepsilon} + \frac{\varepsilon}{2}, & 0 \leq s \leq \varepsilon \\ s, & \varepsilon \leq s \leq \frac{1}{\varepsilon} \\ \frac{\varepsilon s^2}{2} + \frac{1}{2\varepsilon}, & s \geq \frac{1}{\varepsilon} \end{cases} \tag{8}$$

This leads to a family of smooth functionals:

$$E_\varepsilon(u) = \begin{cases} \frac{1}{2} \int_\Omega |Ru - u_0|^2 dx + \lambda \int_\Omega \phi_\varepsilon(|\nabla u|) dx, & u \in W^{1,2}(\Omega) \\ +\infty, & \text{otherwise} \end{cases} \tag{9}$$

These functionals are smooth, which guarantees that each $E_\varepsilon$ has a unique minimizer $u_\varepsilon$ in the Sobolev space $W^{1,2}(\Omega)$.

As $\varepsilon$ decreases, the sequence $\{E_\varepsilon\}$ converges pointwise to $\bar{E}$ on $W^{1,2}(\Omega)$ (and remains infinite elsewhere). More importantly, this sequence $\Gamma$-converges to the extended functional on $BV(\Omega)$ (infinite outside $BV$). Combined with the equi-coercivity of $\{E_\varepsilon\}$, we obtain a strong result: the minimizers $u_\varepsilon$ converge strongly in $L^1(\Omega)$ to the minimizer of $\bar{E}$.

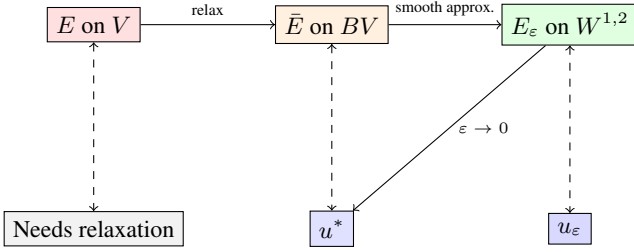

Figure 1: From non-reflexive $V$ to relaxed $BV$ to smooth $W^{1,2}$: the theoretical path enabling neural network solutions for TV denoising.

### 3.4 SOLUTION APPROACHES FOR THE SMOOTHED FUNCTIONAL

The smooth approximations $E_\varepsilon$ enable various numerical solution strategies. Classical approaches include the primal-dual. These methods discretize the domain and solve the resulting finite-dimensional optimization problems. We propose a different approach by parameterizing solutions through neural networks, leveraging physics-informed neural networks Raissi et al. (2019) and operator learning Lu et al. (2021). This continuous parameterization avoids explicit discretization, provides solutions defined everywhere on $\Omega$, and naturally extends to high-dimensional problems.

## 4 METHOD

### 4.1 NEURAL NETWORK PARAMETERIZATION FOR TV MINIMIZATION

We parameterize the solution using a neural network $\mathcal{N}_\theta : \Omega \to \mathbb{R}$ with parameters $\theta \in \mathbb{R}^p$. For each $\varepsilon > 0$, we seek $\theta^*$ such that $\mathcal{N}_{\theta_\varepsilon^*}$ approximates the minimizer $u_\varepsilon^* \in W^{1,2}(\Omega)$ of the smoothed functional $E_\varepsilon$.

### 4.1.1 ACTIVATION FUNCTIONS AND REGULARITY

The selection of smooth activation functions $\sigma \in C^{\infty}(\mathbb{R})$ (tanh, SiLU, softplus) is fundamental to our approach. For a network with $L$ layers:

$$\mathcal{N}_{\theta}(\mathbf{x}) = W_L \sigma(W_{L-1} \sigma(\cdots \sigma(W_1 \mathbf{x} + b_1) \cdots) + b_{L-1}) + b_L$$

where $W_i \in \mathbb{R}^{n_i \times n_{i-1}}$ and $b_i \in \mathbb{R}^{n_i}$.

For any fixed parameter configuration $\theta$, the network $\mathcal{N}_{\theta}$ belongs to $W^{1,2}(\Omega)$. Since $\sigma \in C^{\infty}(\mathbb{R})$ and compositions of smooth functions remain smooth, we have $\mathcal{N}_{\theta} \in C^{\infty}(\Omega)$. On bounded domains $\Omega \subset \mathbb{R}^d$, smooth functions with all their derivatives bounded automatically belong to Sobolev spaces, thus $C^{\infty}(\Omega) \subset W^{k,p}(\Omega)$ for all $k \geq 0$ and $1 \leq p \leq \infty$. In particular, $\mathcal{N}_{\theta} \in W^{1,2}(\Omega)$ for any $\theta \in \mathbb{R}^p$.

Networks with smooth activations approximate functions in the Sobolev norm, not just pointwise. On bounded domains $\Omega \subset \mathbb{R}^d$, the universal approximation theorem for smooth activations Hornik et al. (1989) guarantees simultaneous approximation of both function values and derivatives. When $\mathcal{N}_{\theta}$ approximates $f$, it also approximates $\nabla f$, ensuring convergence in the full $W^{1,2}$ norm $\|u\|_{W^{1,2}}^2 = \|u\|_{L^2}^2 + \|\nabla u\|_{L^2}^2$.

Combining this with the Meyers-Serrin theorem **??**, which states that $C^{\infty}(\Omega) \cap W^{1,2}(\Omega)$ is dense in $W^{1,2}(\Omega)$ for bounded $\Omega$, we obtain that theoretically our parameterization can approximate any minimizer $u_{\varepsilon}^* \in W^{1,2}(\Omega)$ arbitrarily well in both function values and gradients.

### 4.1.2 FUNCTIONAL CONSTRAINTS AND SOLUTION PROPERTIES

This guaranteed membership $\mathcal{N}_{\theta} \in W^{1,2}(\Omega)$ for all $\theta$ serves two critical purposes:

**(i) Constraint Satisfaction and Solution Regularity:** The smoothed functional (equation 9) is defined as:

$$E_{\varepsilon}(u) = \begin{cases} \frac{1}{2}\|Ru - u_0\|_{L^2}^2 + \lambda \int_{\Omega} \phi_{\varepsilon}(|\nabla u|)dx & \text{if } u \in W^{1,2}(\Omega) \\ +\infty & \text{otherwise} \end{cases}$$

Since $\mathcal{N}_{\theta} \in W^{1,2}(\Omega)$ for any $\theta \in \mathbb{R}^p$, our neural network parameterization automatically satisfies the constraint of $E_{\varepsilon}$. Further more, this guarantees that our final solution $\mathcal{N}_{\theta^*}$ belongs to $W^{1,2}(\Omega)$, the correct function space for the variational problem. This is not trivial: the constraint $u \in W^{1,2}(\Omega)$ is essential for the mathematical well-posedness of the TV regularization framework, and our smooth activation functions ensure this constraint is satisfied by construction rather than requiring explicit enforcement during optimization.

**(ii) Well-Defined Physics Constraints:** The Euler-Lagrange equation for $E_{\varepsilon}$ is:

$$R^*(Ru - u_0) - \lambda \nabla \cdot \left( \phi_{\varepsilon}'(|\nabla u|)\frac{\nabla u}{|\nabla u|} \right) = 0$$

With $\mathcal{N}_{\theta} \in C^{\infty}(\Omega)$, this divergence exists and can be computed via automatic differentiation.

## 4.2 PHYSICS-INFORMED TRAINING FOR SINGLE IMAGES

Having established that neural networks with smooth activations provide the necessary $W^{1,2}(\Omega)$ regularity and approximation capabilities, we now detail how we leverage these properties to solve the TV minimization problem through physics-informed training.

### 4.2.1 FROM THEORY TO IMPLEMENTATION

We solve the sequence of smoothed problems $\{E_{\varepsilon}\}$ to approximate the minimizer of the original TV functional $\bar{E}$, exploiting the theoretical convergence $u_{\varepsilon} \to u^*$ as $\varepsilon \to 0$ established in Section 3. The neural network $\mathcal{N}_{\theta} : \Omega \to \mathbb{R}$ directly maps continuous spatial coordinates to function values, transforming the non-differentiable TV minimization into a sequence of smooth optimization problems that standard gradient-based methods can handle.

### 4.2.2 CONTINUOUS REPRESENTATION OF DISCRETE DATA

For single-instance denoising, we construct the training set directly from the noisy image $u_0$. Given discrete data on an $n_1 \times n_2 \times \ldots \times n_d$ grid, we establish a continuous representation by mapping each grid point to coordinates in $[0,1]^d$. Specifically, grid indices $(i_1, i_2, \ldots, i_d)$ map to:

$$\mathbf{x} = \left( \frac{i_1 - 1}{n_1 - 1}, \frac{i_2 - 1}{n_2 - 1}, \ldots, \frac{i_d - 1}{n_d - 1} \right) \in [0,1]^d \tag{10}$$

The training dataset consists of coordinate-value pairs $\{(\mathbf{x}_j, u_0(\mathbf{x}_j))\}_{j=1}^{\prod_{k=1}^d n_k}$. The network $\mathcal{N}_\theta : [0,1]^d \to \mathbb{R}$ takes spatial coordinates as input and outputs the denoised intensity. For all experiments, we synthesize training data by adding Gaussian noise to clean images: $u_0 = u_{\text{clean}} + \eta$ where $\eta \sim \mathcal{N}(0, \sigma^2)$. This controlled setup allows quantitative evaluation against ground truth while demonstrating the unsupervised nature of our method, which only uses $u_0$ during training.

### 4.2.3 PHYSICS-INFORMED LOSS AND TRAINING STRATEGY

For each fixed $\varepsilon$, we enforce the Euler-Lagrange equation by minimizing the residual:

$$\mathcal{L}_{\text{EL}} = \left\| R^*(R\mathcal{N}_\theta - u_0) - \lambda \nabla \cdot \left( \phi_\varepsilon'(|\nabla \mathcal{N}_\theta|) \frac{\nabla \mathcal{N}_\theta}{|\nabla \mathcal{N}_\theta|} \right) \right\|_{L^2(\Omega)}^2 \tag{11}$$

We enforce homogeneous Neumann boundary conditions through:

$$\mathcal{L}_{\text{BC}} = \left\| \frac{\partial \mathcal{N}_\theta}{\partial n} \right\|_{L^2(\partial \Omega)}^2 \tag{12}$$

The total loss combines both terms: $\mathcal{L} = \mathcal{L}_{\text{EL}} + \lambda_{\text{BC}} \mathcal{L}_{\text{BC}}$.

The key innovation is our progressive $\varepsilon$-reduction strategy. Starting from a large $\varepsilon_0$ where the problem is highly smooth, we solve a sequence of increasingly less regularized problems. After convergence at each $\varepsilon$, we reduce it by a factor $\alpha \in (0,1)$ and continue training with warm-starting. The complete procedure is formalized in Algorithm 1.

---

**Algorithm 1** PINN Training for TV Minimization

---

1: **Input:** Network $\mathcal{N}_\theta$, initial $\varepsilon_0 > 0$, decay $\alpha \in (0,1)$, tolerance $\tau$
2: **Output:** Network $\mathcal{N}_{\theta^*}$ approximating TV minimizer
3: $\varepsilon \leftarrow \varepsilon_0$
4: **while** $\varepsilon > \tau$ **do**
5:     Sample coordinate batch $\{\mathbf{x}_j\}_{j=1}^M \subset [0,1]^d$
6:     **repeat**
7:         Evaluate $u_0(\mathbf{x}_j)$ via interpolation from discrete data
8:         Compute residual: $r_j = R^*(R\mathcal{N}_\theta(\mathbf{x}_j) - u_0(\mathbf{x}_j)) - \lambda \mathcal{D}_\varepsilon[\mathcal{N}_\theta](\mathbf{x}_j)$
9:         Update $\theta$ to minimize $\mathcal{L} = \frac{1}{M} \sum_j |r_j|^2 + \lambda_{\text{BC}} \mathcal{L}_{\text{BC}}$
10:    **until** convergence
11:    $\varepsilon \leftarrow \alpha \cdot \varepsilon$
12:    Warm start: retain $\theta$ for next iteration
13: **end while**
14: **return** $\mathcal{N}_{\theta^*}$

---

This approach combines the theoretical guarantees from our neural network parameterization with practical computational efficiency. Implementation details are provided in Appendix **??**. While effective for single images, practical applications require processing multiple images efficiently, motivating our operator learning extension.

### 4.3 OPERATOR LEARNING FOR MULTIPLE IMAGES

While PINNs solve individual denoising instances, practical applications demand processing multiple images efficiently. We extend our approach to learn the TV denoising operator that generalizes across a distribution of noisy inputs.

#### 4.3.1 DEEPONET ARCHITECTURE FOR TV REGULARIZATION

We employ the Deep Operator Network (DeepONet) framework Lu et al. (2021), which learns mappings between function spaces through a branch-trunk decomposition:

$$G_\theta(u_0)(\mathbf{x}) = \sum_{i=1}^{p} B_i(u_0) \cdot T_i(\mathbf{x}) \tag{13}$$

The branch network $B_\theta : \mathbb{R}^m \to \mathbb{R}^p$ encodes the noisy input $u_0$ sampled at $m$ fixed sensor locations. The trunk network $T_\theta : [0,1]^d \to \mathbb{R}^p$ maps spatial coordinates to features. Crucially, we use smooth activations (SiLU) in the trunk to ensure $T \in C^\infty([0,1]^d)$, maintaining the $W^{1,2}$ regularity required for the Euler-Lagrange equation. The branch can use non-smooth activations (ReLU) since derivatives are only computed with respect to trunk coordinates.

#### 4.3.2 TRAINING DATASET CONSTRUCTION

For operator learning, we prepare $N$ noisy instances $\{u_0^{(k)}\}_{k=1}^N$. Each instance is normalized to $[0,1]$ intensity range, consistent with our PINN approach. The operator conceptually maps triplets $(u_0, \mathbf{x}_{in}, \mathbf{x}_{out})$ where $u_0$ is the normalized noisy input image, $\mathbf{x}_{in}$ are the sensor locations where we sample the input, and $\mathbf{x}_{out}$ are the coordinates where we evaluate the denoised output.

However, the unsupervised nature of our approach imposes a critical constraint. To evaluate the fidelity term in the Euler-Lagrange equation, we need to compute $G_\theta(u_0)(\mathbf{x}) - u_0(\mathbf{x})$ at the same spatial location $\mathbf{x}$. This requirement collapses the triplet to a pair by enforcing $\mathbf{x}_{in} = \mathbf{x}_{out}$.

Specifically, for each training batch we sample coordinates $\{\mathbf{x}_j\}_{j=1}^M \subset [0,1]^d$. The branch network receives the normalized noisy image $u_0$ evaluated at these coordinates as $\{u_0(\mathbf{x}_j)\}_{j=1}^M$, while the trunk network receives the same coordinates $\{\mathbf{x}_j\}_{j=1}^M$. The loss function then compares $G_\theta(u_0)(\mathbf{x}_j)$ with $u_0(\mathbf{x}_j)$ at identical locations. This alignment between sensor and evaluation locations, combined with intensity normalization, enables unsupervised learning without clean ground truth. For operator learning, we prepare $N$ noisy instances $\{u_0^{(k)}\}_{k=1}^N$ by adding Gaussian noise from the same distribution to clean images: $u_0^{(k)} = u_{\text{clean}}^{(k)} + \eta^{(k)}$ where $\eta^{(k)} \sim \mathcal{N}(0, \sigma^2)$ with fixed $\sigma$ across the dataset. Each instance is then normalized to $[0,1]$ intensity range. This consistency in noise characteristics enables the operator to learn a robust denoising mapping.

#### 4.3.3 PHYSICS-INFORMED OPERATOR TRAINING

We apply the same $\varepsilon$-reduction strategy to the operator setting. For each fixed $\varepsilon$, we minimize the physics residual across all instances:

$$\mathcal{L}_{\text{EL}} = \frac{1}{NM} \sum_{i=1}^{N} \sum_{j=1}^{M} \left\| R^*(RG_\theta(u_0^{(i)})(\mathbf{x}_j) - u_0^{(i)}(\mathbf{x}_j)) - \lambda \mathcal{D}_\varepsilon[G_\theta(u_0^{(i)})](\mathbf{x}_j) \right\|^2 \tag{14}$$

where $\mathcal{D}_\varepsilon$ denotes the divergence term from the Euler-Lagrange equation. Boundary conditions are enforced similarly across all instances. The complete procedure is given in Algorithm 2.

**Algorithm 2** Physics-Informed Operator Learning for TV Denoising

---

1: **Input:** Dataset $\{u_0^{(i)}\}_{i=1}^N$, initial $\varepsilon_0 > 0$, decay $\alpha \in (0,1)$, tolerance $\tau$
2: **Output:** Operator $G_{\theta*}$ approximating TV denoising operator
3: Initialize branch network $B_\theta$ and trunk network $T_\theta$
4: $\varepsilon \leftarrow \varepsilon_0$
5: **while** $\varepsilon > \tau$ **do**
6:     Sample triplet batch $\{(u_0^{(i_k)}, \mathbf{x}_j, u_0^{(i_k)}(\mathbf{x}_j))\}$ for $k = 1, \ldots, N_b$ and $j = 1, \ldots, M$
7:     **repeat**
8:         Compute operator output: $G_\theta(u_0^{(i_k)})(\mathbf{x}_j) = \sum_l B_l(u_0^{(i_k)}) \cdot T_l(\mathbf{x}_j)$
9:         Compute residuals and update $\theta$ to minimize $\mathcal{L}_{\mathrm{EL}} + \lambda_{\mathrm{BC}}\mathcal{L}_{\mathrm{BC}}$
10:   **until** convergence
11:     $\varepsilon \leftarrow \alpha \cdot \varepsilon$
12:     Warm start: retain $\theta$ for next iteration
13: **end while**
14: **return** $G_{\theta*}$

---

Implementation details are provided in Appendix **??**.

## 5 EXPERIMENTS AND RESULTS

### 5.1 EXPERIMENTAL SETUP

We evaluate our methods on 3D imaging tasks. We compare against the Chambolle primal algorithm Chambolle (2004) as it provably converges to the same TV minimizer we approximate. Both methods solve identical variational problems, isolating the effect of neural parameterization versus classical discretization. For fair comparison, we optimize the baseline extensively. For each test case, we evaluate Chambolle with 10 different regularization parameters $\lambda$ logarithmically spaced around the noise level, reporting the best SSIM resuls for optimal performance, as $\lambda$ controls the trade-off between data fidelity and TV regularization.

We evaluate using Peak Signal-to-Noise Ratio (PSNR), Structural Similarity Index (SSIM), and RMSE.

### 5.2 3D VOLUME DENOISING WITH PINN

We demonstrate our PINN approach on a synthetic 3D heart cone phantom of size $64^3$ voxels. The phantom consists of a cone-shaped region with intensity 1 inside and 0 outside, creating a pure jump discontinuity at the cone boundary. This binary structure specifically tests TV regularization's core strength: preserving sharp edges defined by the jump set $S_u$ in the BV decomposition. We corrupt this binary phantom with Gaussian noise ($\sigma = 0.2$), a challenging noise level that severely obscures the sharp boundary.

|  | RMSE ↓ | | PSNR ↑ | | SSIM ↑ | |
|---|---|---|---|---|---|---|
|  | Chambolle TV | Ours | Chambolle TV | Ours | Chambolle TV | Ours |
| Full volume | 0.172 | **0.154** | 15.29 | **16.27** | 0.875 | **0.921** |
| Slice 1 | 0.188 | **0.183** | 14.50 | **14.70** | 0.860 | **0.900** |
| Slice 49 | 0.146 | **0.136** | 16.70 | **17.33** | 0.905 | **0.931** |

Table 1: Quantitative results for 3D heart cone denoising. TV denotes Chambolle with optimal $\lambda$ selected from 10 candidates.

Our PINN method consistently outperforms the optimally-tuned Chambolle baseline across all metrics. The full volume shows a 10.5% reduction in RMSE and nearly 1 dB improvement in PSNR. The SSIM improvement of 0.046 is particularly significant for this binary phantom, as it indicates better structural preservation of the cone geometry. The slice-wise analysis reveals interesting patterns. Slice 1, with a larger cross-section near the cone base, shows our method achieving SSIM

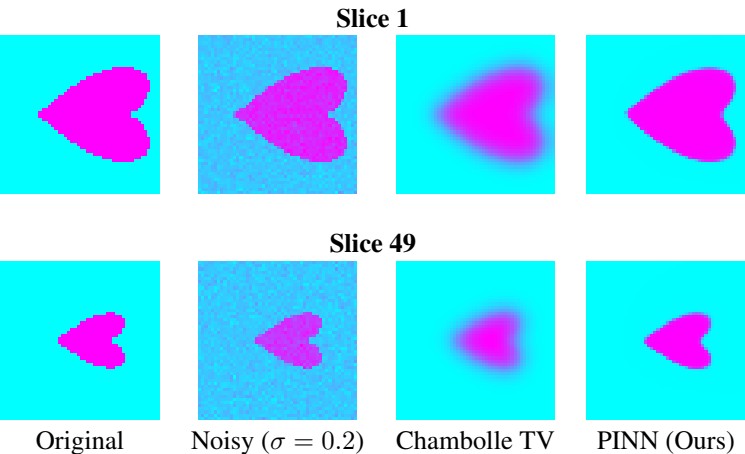

Figure 2: 3D heart cone denoising showing two representative slices.

of 0.900 versus 0.860 for TV, a 4.7% improvement. Slice 49, near the apex with minimal cross-section, demonstrates even stronger performance with SSIM of 0.931 versus 0.905. The superior performance on both slices, particularly on the geometrically simpler Slice 49, suggests our continuous representation effectively captures the jump discontinuity regardless of boundary complexity. as seen in Figure 2 Figure 2 visualizes these quantitative gains. While both methods remove noise, our PINN produces cleaner boundaries. The continuous neural representation avoids the staircase artifacts visible in the TV solution, particularly evident along the cone edges.

## 5.3 2D OPERATOR LEARNING ON CHESTMNIST

We evaluate on 2D medical images from the ChestMNIST dataset. We train a single DeepONet on 100 chest X-ray images, Each volume has dimensions $64 \times 64$ pixels testing the operator's ability to generalize across varying structures.

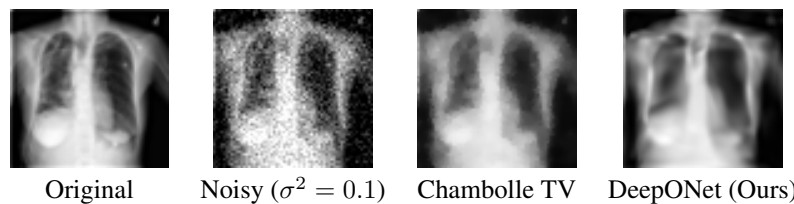

Figure 3: ChestMNIST denoising results. Our operator preserves anatomical structures while effectively removing noise.

| Method | RMSE ↓ | PSNR ↑ | SSIM ↑ |
|---|---|---|---|
| Chambolle TV | 0.00020 | 26.2 | 0.886 |
| DeepONet (Ours) | **0.00019** | **27.3** | **0.910** |

Table 2: Quantitative results on ChestMNIST with $\sigma^2 = 0.1$. TV uses optimal $\lambda$ from 10 candidates.

Our operator achieves consistent improvements: 1.1 dB PSNR gain and 0.024 SSIM improvement over the Chambolle TV. The improvements demonstrate that our physics-informed operator successfully generalizes to 2D. Figure 3 shows that our method better preserves fine details while avoiding the over-smoothing visible in the TV solution. Additional experiment with higher noise levels ($\sigma^2 = 0.3$) are provided in Appendix **??**.

We evaluate our physics-informed DeepONet on the OrganMNIST3D dataset, learning a single operator to denoise 100 different organ volumes. Each volume has dimensions $64 \times 64 \times 12$ voxels, representing various organ types with distinct anatomical structures. We corrupt all volumes with Gaussian noise ($\sigma = 0.1$) to test the operator's ability to generalize across different anatomies. The DeepONet architecture employs a branch network (4-layer CNN followed by MLP with ReLU activations, 7500 hidden units) to encode noisy images at fixed sensor locations, and a trunk network (4-layer MLP with SiLU activations, 5500 hidden units) to encode spatial coordinates. The output dimension of 7500 combines as $G_\theta(u_0)(\mathbf{x}) = \sum_{i=1}^{7500} B_i(u_0) \cdot T_i(\mathbf{x})$.

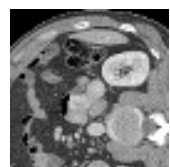 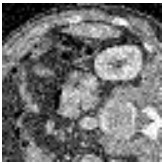 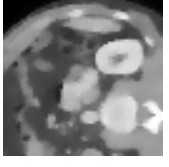 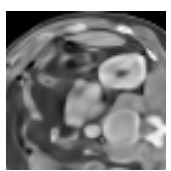

Original          Noisy ($\sigma = 0.1$)          Chambolle TV          DeepONet (Ours)

Figure 4: Representative slice from OrganMNIST3D volume ($64 \times 64 \times 12$). Our operator method preserves organ boundaries while reducing noise.

| Method | PSNR $\uparrow$ | SSIM $\uparrow$ | RMSE $\downarrow$ |
|---|---|---|---|
| Chambolle TV | 21.10 | 0.767 | 0.00776 |
| DeepONet (Ours) | **21.55** | **0.836** | **0.00700** |

Table 3: Quantitative results for the representative OrganMNIST3D slice shown in Figure 4.

Our DeepONet operator consistently outperforms the optimally-tuned Chambolle baseline. The RMSE reduction of 9.8% and PSNR improvement of 0.45 dB demonstrate effective noise removal across diverse organ types. The SSIM improvement of 0.069 indicating better preservation of anatomical structures despite the operator being trained on multiple organ types simultaneously.

Figure 4 shows the qualitative differences. While both methods remove noise, our operator better preserves fine anatomical details and organ boundaries. The continuous neural representation avoids the over-smoothing artifacts visible in the TV solution, maintaining internal organ texture while still achieving effective denoising.

## 6 CONCLUSION

We presented a physics-informed neural network approach for total variation denoising that bridges variational methods with deep learning. By solving the sequence of smoothed problems $E_\varepsilon$ through progressive reduction, our method approximates the TV solution while maintaining theoretical guarantees through the $\Gamma$-convergence framework. Our key contributions include: (1) demonstrating that coordinate-based neural networks can effectively solve the TV problem in both 2D and 3D, achieving consistent improvements over classical methods despite using smooth activations to approximate discontinuous functions; (2) extending to operator learning for processing multiple images efficiently. **Limitations and Future Work:** We lack explicit approximation rates for neural networks in Sobolev spaces while bounds exist for linear PDEs Ben-Shaul et al. (2023), extending to nonlinear problems remains open. The precise function space of trained PINNs requires characterization. Future directions include establishing convergence rates, exploring direct energy minimization versus Euler-Lagrange residuals, and incorporating attention mechanisms for improved operator learning. Additionally, validation on real medical data with acquisition-specific noise models would strengthen practical applicability. Despite these theoretical gaps, our empirical results demonstrate that physics-informed approaches offer a promising direction for combining the mathematical rigor of variational methods with the computational advantages of neural networks, particularly for high-dimensional imaging problems where classical methods face computational constraints.

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
