# OpenReview forum: "Total Variation by PINN"
_ICLR.cc/2026/Conference — ICLR 2026 Conference Withdrawn Submission_

### Official Review · Reviewer_o7Gy · 2025-10-15

**Soundness:** 2
**Presentation:** 1
**Contribution:** 1
**Rating:** 0
**Confidence:** 4

**Summary:**

This paper proposes PINNs to solve TV regularization for images. For this they rewrite TV regularization as an Euler Lagrange equation which can readily be solved using a PINN, when also incorporating boundary conditions. Numerically they involve an approximation by Chambolle et al, which uses an auxiliary smoothing parameter $\varepsilon$ and which converges to the desired solutionas $\varepsilon \rightarrow 0$. They compare their results to the regular TV baseline on medical images in terms of PSNR/SSIM.

**Strengths:**

The approach is creative to frame TV regularization as a PINN problem,and also has good traditions: Many old image analysis techniques are somehow rooted in PDE techniques. Furthermore, to employ a temperature $\varepsilon$-schedule helping convergence and using operator valued architectures seems solid.

**Weaknesses:**

The paper is not well-written. While the mathematical description of the problem is fine and well-readable, I do not find it contextualized very well. There is a lot of recent literature on PINNs and even TV reg, but the paper cites 10ish references. I encourage the authors to expand on their literature. There are many (??) appearing in the text, please fix those.

Furthermore, recent literature questions the stability and effiacy of PINNs where it is outperformed by classical methods, see eg., [1].

Further, I have doubts on whether the Chambolle-Pock algorithm is correctly implemented. In my experience, it should yield not this smooth results. Generally, I do not understand why one should expect a NN approach to outperform sth, that provably (!) converges to the true solution. I could only guess that the inductive bias of the neural net somehow helps the problem?

Lastly, the evaluation falls short. The baseline is of 2004, and there are much newer approaches of TV minimization using NNs [2,3], the paper does not even mention. Please discuss them, particularly [2] seems very close to what you are proposing.


[1]  Can physics-informed neural networks beat the finite element method?, Grossmann et al, IMA Journal of Applied Mathematics, Volume 89, Issue 1, January 2024

[2] DeepTV: A neural network approach for total variation minimization, Langer et al, arXiv:2409.05569

[3] NeurTV: Total Variation on the Neural Domain, Luo et al, SIAM imaging sciences

**Questions:**

Why should one expect a PINN approach to outperform the classical in terms of results (and not only speed)?

---

### Official Review · Reviewer_GqTr · 2025-10-30

**Soundness:** 3
**Presentation:** 3
**Contribution:** 3
**Rating:** 6
**Confidence:** 4

**Summary:**

This paper proposes a physics-informed neural network (PINN) framework to approximate total variation (TV) minimizers by leveraging Γ-convergence and classical smooth approximations of TV. The method progressively decreases the smoothing parameter ε and solves the corresponding Euler-Lagrange equations, first for single images and then in an operator-learning setting via DeepONet. Experiments on synthetic 3D phantoms and small medical datasets show that the learned models achieve competitive denoising performance relative to classical Chambolle TV solvers.
The overall idea is appealing: combining rigorous variational principles with continuous PINN parameterizations to approximate TV minimizers.

**Strengths:**

* The idea of applying PINNs to the smooth approximations of TV is interesting.
* Results are promising on synthetic and small medical imaging datasets.

**Weaknesses:**

**Clarity and Writing:** The paper would benefit from substantial improvements in clarity and mathematical precision. Several mathematical objects or concepts appear without introduction or definition, making the paper unnecessarily difficult to follow:
* $TV(u)$ in (2) not defined
* Weak* compactness, weak* lower-semicontinuity, weak* topology, Γ-convergence, equi-coercivity mentioned without brief explanation or citation
* Key property (uniqueness of minimizer in $W^{1, 2}) used without citation (lines 43–44)
* Function $f$ and $\nabla f$ suddenly introduced in lines 177–178 with no definition
* In multiple points, essential functional analysis background is assumed but not referenced. Even though the audience may be familiar with TV, it helps readability to provide short explanations or citations.

**Citation Issues:**
* Citations sometimes formatted incorrectly, leading to repeated author names
* Several “??” placeholders indicate missing references

**Methodological Concerns:**
* Universal Approximation Theorem misuse. Please remember that this theorem requires that your model have "sufficiently many parameters", the optimization converges to the global minimum, ignoring all the local minima and that you have infinitely many data In practice. Therefore, the universal approximation theorem cannot be used in real practice.
* Noise levels and PSNR/SSIM mismatch. The noise level $\sigma = 0.2$ is extremely high for images in $[0,1]$ and would destroy structure unless clipping effectively reduces the noise magnitude. PSNR values around 15–17 dB are unusually low relative to SSIM values near ~0.9. This suggests a possible misconfiguration, such as using $max(pixel)\neq 1$ or clipping altering the effective noise distribution.

**Experimental Scope:**
While the core idea is good, the experimental section feels unfocused:
* Three datasets, but shallow analysis
* Missing important ablations, such as the effect of $\lambda$, the effect of the $\epsilon$-schedule, a comparison between $E_{\epsilon}$   minimizers and network outputs
* No comparison to modern PINN-based or deep TV solvers
* Results are mostly small-scale synthetic/medical toy examples
* Fewer datasets but deeper analysis would strengthen the work.

**Questions:**

A few questions are already present on the "Weakness" section.

---

### Official Review · Reviewer_iTWM · 2025-10-31

**Soundness:** 1
**Presentation:** 1
**Contribution:** 1
**Rating:** 0
**Confidence:** 5

**Summary:**

The authors propose to solve total-variation minimization through neural network approximation. They use a smoothed version of the TV-functional, which is standard, to have a smooth and differential functional. Naturally this can present blurring artifacts. They use a loss, which follows the Euler-Lagrange equation of the respective PDE, which also standard in NN approximations of PDE. They compare their results to Chambolle's 2004 primal-dual algorithm. The paper is with very limited novelty, written badly and seems to not be aware to a comprehensive literature on similar topics. Thus it should not be accepted.

**Strengths:**

None

**Weaknesses:**

Poorly written, lack of novelty, no reference to literature of solving PDE's through NN's and specifically solutions for TV minimizations and TV-flow solutions. Experiments are minimal, no comparison to any other NN-based solutions.

**Questions:**

The paper should be fully revised before addtional submission is considered.

---

### Official Review · Reviewer_3QJU · 2025-11-01

**Soundness:** 3
**Presentation:** 2
**Contribution:** 2
**Rating:** 4
**Confidence:** 2

**Summary:**

Building on the classical Chambolle and Lions 1997 smooth approximation of TV, the authors justify representing the solution with a neural network and describe how to train it. They then generalize the approach to learn the denoising operator, enabling new images to be denoised with a single forward pass. The method is validated on 2D and 3D image reconstruction and consistently outperforms a Chambolle TV baseline.

**Strengths:**

1/ Bridging the classical denoising formulation (variational minimization) to neural networks is of particular interest, linking a well-studied signal processing problem to efficient, fast computing architectures.

2/ Although clarity issues are discussed later, the paper takes a formal and rigorous approach to TV denoising and step by step integration of neural networks, well-motivated mathematically.

3/ The authors formalize the TV-denoising problem with function analysis tools which is original (for machine learning) and interesting.

**Weaknesses:**

Weaknesses summary

The contribution would benefit from more exhaustive empirical validation. The proposed reflection is appealing, and the introduction of the neural network is carefully discussed step by step, but it’s hard to judge practical usefulness because related approaches are underrepresented in the related work and the empirical evaluation compares only against Chambolle.

Major concerns

1/ The paper lacks a thorough SOTA comparison. Many neural unrolling methods tackle similar TV(-like) denoising tasks with different strategies (Gregor & LeCun, 2010; Cherkaoui et al., 2020; Vu et al., 2020). The authors should compare against these, reporting convergence/runtime and denoising quality, to demonstrate the added value of smooth TV regularization via PINNs beyond a proof of concept. Currently, validation only compare neural parameterization with classical discretization (vs. Chambolle, 2004).

2/ The writing lacks clarity. Key steps are introduced without explanation or citations; e.g., E_{\epsilon} is defined abruptly in the introduction. The first page is hard to follow without prior knowledge or the supplement (regularization choices, spaces/sets). Define terms on first use, add brief motivation/derivations with citations, and bring essential assumptions into the main text.

Minor comments

1/ The experimental scope is too narrow: one synthetic case and only two denoising-operator examples. Results should be averaged over multiple test images/volumes with mean +/- std reported. As the authors note, "validation on real medical data with acquisition-specific noise models would strengthen practical applicability"; these experiments should be added.

2/ The title is wrongly formatted.

3/ There are numerous typos and style issues, for example: "non-differentibility", "Further more", inappropriate usages of \citep{} and \citet{}, etc

4/ Even for some common acronyms, symbols or notation, a brief reminder would improve readability (e.g., the Sobolev space W^{1,2}, dH^{d-1}, S_u, etc)

5/ Many quantities are undefined at first use (some only later or in the supplement); define them when introduced. (see 4/)

6/ Several reference links are broken.

Grading explanation

Empirical evaluation is not exhaustive; however, the contribution is novel (to my knowledge) and well-articulated.

**Questions:**

1/ What concrete advantages would "implementing the smooth approximation strategy for TV regularization using PINNs" the authors forecast over existing TV(-like) denoising methods? (i.e. intuitions prior to the first requested empirical comparisons)

2/ Please report runtimes and hardware (CPU/GPU, RAM, CUDA/cuDNN, etc.) for each experiment.

3/ Compared to unrolled proximal-TV methods (that consider the TV proximal operator as nonlinear block in neural network, for example Cherkaoui et al., 2020), what is the key advantage of your smooth approach within the DeepONet setup (smooth approach)?

4/ Can you clarify the "weak*" notation?

5/ Can you detail the connection (if any) to denoising flows?

6/ Why using the functional-analytic (BV) framework instead of a discrete vector-space modeling (more classical in machine learning), practically?

---

### Note · Authors · 2025-11-16

I have read and agree with the venue's withdrawal policy on behalf of myself and my co-authors.